# Nodal patterning without Lefty inhibitory feedback is functional but fragile

Katherine W Rogers[1†‡*], Nathan D Lord[1†], James A Gagnon[1], Andrea Pauli[1§], Steven Zimmerman[1], Deniz C Aksel[2], Deepak Reyon[3,4,5,6#], Shengdar Q Tsai[3,4,5,6¶], J Keith Joung[3,4,5,6], Alexander F Schier[1,7,8,9,10*]

[1]Department of Molecular and Cellular Biology, Harvard University, Cambridge, United States; [2]Program in Biophysics, Harvard Medical School, Boston, United States; [3]Molecular Pathology Unit, Massachusetts General Hospital, Charlestown, United States; [4]Department of Pathology, Harvard Medical School, Boston, United States; [5]Center for Computational and Integrative Biology, Massachusetts General Hospital, Boston, United States; [6]Center for Cancer Research, Massachusetts General Hospital, Charlestown, United States; [7]Broad Institute of MIT and Harvard University, Cambridge, United States; [8]Center for Brain Science, Harvard University, Cambridge, United States; [9]Harvard Stem Cell Institute, Harvard University, Cambridge, United States; [10]Center for Systems Biology, Harvard University, Cambridge, United States

*For correspondence:
katherine.rogers@tuebingen.mpg.de (KWR);
schier@fas.harvard.edu (AFS)

†These authors contributed equally to this work

Present address: ‡Systems Biology of Development Group, Friedrich Miescher Laboratory of the Max Planck Society, Tübingen, Germany; §The Research Institute of Molecular Pathology, Vienna, Austria; #Editas Medicine, Inc., Cambridge, United States; ¶St. Jude Children's Research Hospital, Memphis, United States

**Abstract** Developmental signaling pathways often activate their own inhibitors. Such inhibitory feedback has been suggested to restrict the spatial and temporal extent of signaling or mitigate signaling fluctuations, but these models are difficult to rigorously test. Here, we determine whether the ability of the mesendoderm inducer Nodal to activate its inhibitor Lefty is required for development. We find that zebrafish *lefty* mutants exhibit excess Nodal signaling and increased specification of mesendoderm, resulting in embryonic lethality. Strikingly, development can be fully restored without feedback: Lethal patterning defects in *lefty* mutants can be rescued by ectopic expression of *lefty* far from its normal expression domain or by spatially and temporally uniform exposure to a Nodal inhibitor drug. While drug-treated mutants are less tolerant of mild perturbations to Nodal signaling levels than wild type embryos, they can develop into healthy adults. These results indicate that patterning without inhibitory feedback is functional but fragile.
DOI: https://doi.org/10.7554/eLife.28785.001

## Introduction

Feedback inhibition is a common feature of developmental pathways across phyla (*Freeman, 2000*; *Freeman and Gurdon, 2002*; *Meinhardt, 2009*; *Piddini and Vincent, 2009*; *Ribes and Briscoe, 2009*; *Rogers and Schier, 2011*). Feedback inhibitors contribute to patterning processes in tissues ranging from the mouse neural tube (*Ribes and Briscoe, 2009*) and the zebrafish hindbrain (*White and Schilling, 2008*), to the *Drosophila* wing (*Gerlitz and Basler, 2002*; *Piddini and Vincent, 2009*; *Zeng et al., 2000*) and eye (*Freeman, 1997*). Consistent with a general requirement for feedback control in development, inactivation of feedback inhibitors often results in disastrous patterning defects. However, directly testing the role of feedback per se has remained challenging: Eliminating inhibitors removes feedback, but it also increases signaling levels. Experiments that decouple inhibitor activation and inhibition while maintaining near-normal signaling levels are therefore required to unambiguously test the role of feedback in developmental patterning.

**eLife digest** During animal development, a single fertilized cell gives rise to different tissues and organs. This 'patterning' process depends on signaling molecules that instruct cells in different positions in the embryo to acquire different identities. To avoid mistakes during patterning, each cell must receive the correct amount of signal at the appropriate time.

In a process called 'inhibitory feedback', a signaling molecule instructs cells to produce molecules that block its own signaling. Although inhibitory feedback is widely used during patterning in organisms ranging from sea urchins to mammals, its exact purpose is often not clear. In part this is because feedback is challenging to experimentally manipulate. Removing the inhibitor disrupts feedback, but also increases signaling. Since the effects of broken feedback and increased signaling are intertwined, any resulting developmental defects do not provide information about what feedback specifically does. In order to examine the role of feedback, it is therefore necessary to disconnect the production of the inhibitor from the signaling process.

In developing embryos, a well-known signaling molecule called Nodal instructs cells to become specific types – for example, a heart or gut cell. Nodal also promotes the production of its inhibitor, Lefty. To understand how this feedback system works, Rogers, Lord et al. first removed Lefty from zebrafish embryos. These embryos had excessive levels of Nodal signaling, did not develop correctly, and could not survive. Bathing the embryos in a drug that inhibits Nodal reduced excess signaling and allowed them to develop successfully. In these drug-treated embryos, inhibitor production is disconnected from the signaling process, allowing the role of feedback to be examined. Drug-treated embryos were less able to tolerate fluctuations in Nodal signaling than normal zebrafish embryos, which could compensate for such disturbances by adjusting Lefty levels.

Overall, it appears that inhibitory feedback in this patterning system is important to compensate for alterations in Nodal signaling, but is not essential for development. Understanding the role of inhibitory feedback will be useful for efforts to grow tissues and organs in the laboratory for clinical use. The results presented by Rogers, Lord et al. also suggest the possibility that drug treatments could be developed to help correct birth defects in the womb.

DOI: https://doi.org/10.7554/eLife.28785.002

Inhibitory feedback has been invoked to explain the sophisticated spatiotemporal control and robust performance observed in patterning circuits (*Ribes and Briscoe, 2009*). Inhibitory feedback has been suggested to turn off pathway activity when it is no longer needed, and to regulate spatial profiles of pathway activation (*Barkai and Shilo, 2009*; *Ben-Zvi et al., 2008*; *Dessaud et al., 2007*; *Freeman, 2000*; *Gerlitz and Basler, 2002*; *Golembo et al., 1996*; *Lecuit and Cohen, 1998*; *Piddini and Vincent, 2009*; *Ribes and Briscoe, 2009*; *Schilling et al., 2012*; *Shiratori and Hamada, 2006*; *van Boxtel et al., 2015*). Additionally, theoretical considerations suggest that negative feedback could enable the embryo to adjust signaling levels in response to unexpected perturbations or biochemical fluctuations (*Barkai and Shilo, 2009*; *Eldar et al., 2003*; *Lander et al., 2009*).

The Nodal/Lefty system has become a paradigm for feedback inhibition in development (*Chen and Schier, 2002*; *Duboc et al., 2008*; *Freeman, 2000*; *Hamada, 2012*; *Kondo and Miura, 2010*; *Meinhardt, 2009*; *Meno et al., 1999*; *Nakamura et al., 2006*; *Rogers and Schier, 2011*; *Schier, 2009*; *Shen, 2007*). Nodal is a TGFβ superfamily ligand that induces the phosphorylation and subsequent nuclear localization of the transcription factor Smad2 in target cells. In early embryos, Nodal signaling induces mesendodermal fates through graded activation of target genes, with higher signaling intensities biasing cells toward endoderm. Nodal signaling intensity is shaped by the interplay between Nodal ligands and Leftys, secreted signaling inhibitors that prevent Nodal from binding to its receptors (*Chen and Shen, 2004*; *Cheng et al., 2004*). Studies of mouse *lefty* mutants and zebrafish *lefty* morphants reveal that loss of inhibition results in expanded domains of Nodal target expression, increased mesendodermal specification, and embryonic lethality (*Agathon et al., 2001*; *Chen and Schier, 2002*; *Feldman et al., 2002*; *Meno et al., 1999*; *van Boxtel et al., 2015*). Antagonism by Lefty is thus important for preventing overactive Nodal signaling during mesendodermal patterning.

Lefty production is coupled to Nodal signaling, forming a negative feedback loop that is conserved from sea urchins to humans. For example, in zebrafish, *lefty1* and *lefty2* are induced by endogenous Nodal signaling at the blastoderm margin, expression of Nodal can drive ectopic *lefty* production, and loss of Nodal signaling abolishes expression of *lefty* (*Figure 1A,B*) (*Meno et al.,*

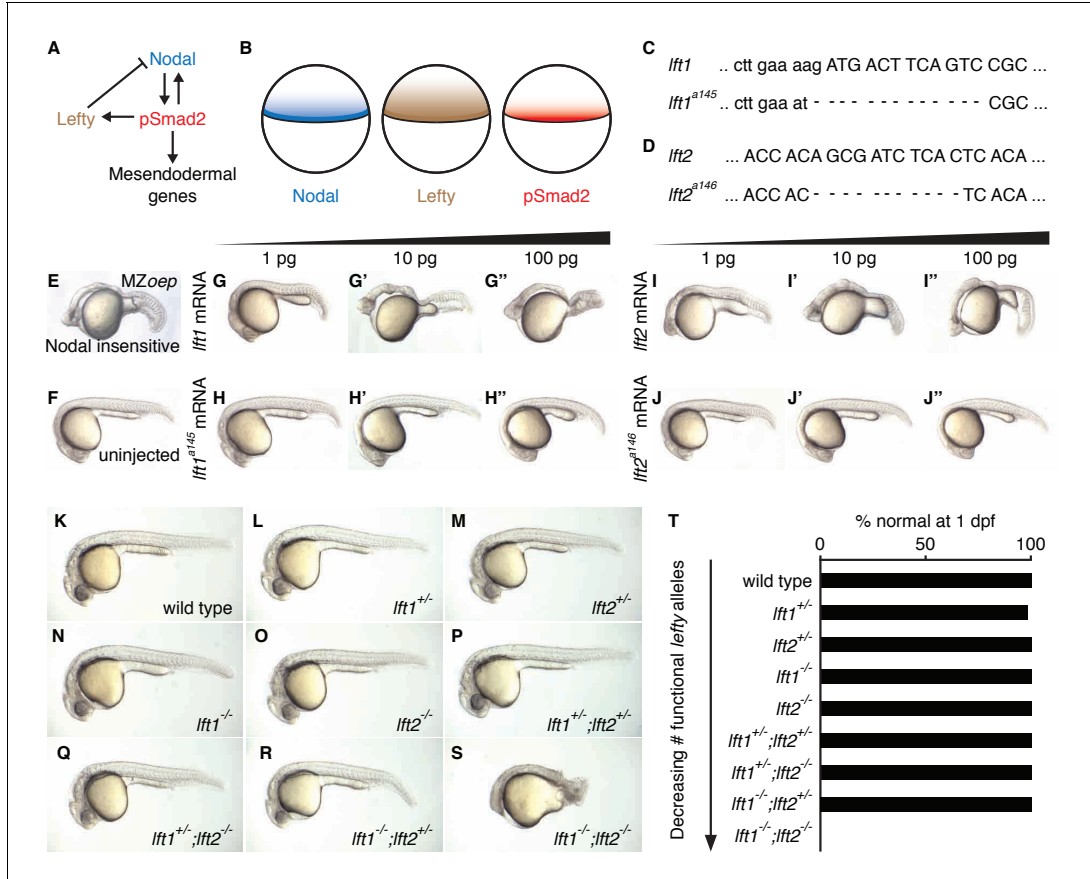

**Figure 1.** Complete Lefty loss causes severe patterning defects. (A) Nodal activates itself, mesendodermal genes, and the secreted feedback inhibitor Lefty by inducing phosphorylation and nuclear translocation of the signal transducer Smad2. (B) Nodal (blue) and Lefty (brown) are expressed in overlapping domains at the margin (black line) in zebrafish embryos, generating a signaling gradient of phosphorylated Smad2 (red). (C) A 13-base-pair deletion at the 5' end of *lft1[a145]* removes the translational start site (TSS) and part of the signal sequence. An alternative TSS 36 bp downstream of the deletion could produce an in-frame protein product, but the first 16 amino acids, and therefore most of the predicted 20-aa signal sequence, would be missing. (D) An 11-base-pair deletion at the 5' end of *lft2[a146]* removes part of the predicted 19-aa signal sequence, causing a frame shift after 36 bp resulting in a stop codon 18 bp later. (E–J'') Testing the activity of *lft* mutant mRNA. All images were obtained at 24 hr post-fertilization (hpf). Wild type embryos at the one-cell stage were injected with 1, 10, and 100 pg wild type *lft1* (G–G'') or *lft1[a145]* (H–H'') mRNA as indicated. Embryos expressing *lft1* mRNA exhibit Nodal loss-of-function phenotypes, similar to maternal-zygotic mutants for the zebrafish EGF-CFC co-receptor *one-eyed pinhead* (*oep*) (E), which are insensitive to Nodal signaling (*Gritsman et al., 1999*). Embryos expressing mutant mRNA do not exhibit Nodal loss-of-function phenotypes. Similar results were obtained for wild type *lft2* (I–I'') and *lft2[a146]* (J–J'') mRNA. (F) Uninjected wild type embryo. (K–S) *lft* mutant phenotypes. All images were obtained at 24–29 hpf. A single functional *lft* allele is sufficient for grossly normal patterning (Q,R). (S) *lft1[-/-]*;*lft2[-/-]* double homozygous mutants have severe patterning defects and lack eyes, heart, and full length tails, and often exhibit excess tissue along the posterior trunk. (T) Percentage of embryos with normal gross morphology at 1 day post-fertilization (dpf). Number of normal/total embryos: wild type = 50/50, *lft1[+/-]* = 49/50, *lft2[+/-]* = 46/46, *lft1[-/-]* = 50/50, *lft2[-/-]* = 46/46, *lft1[+/-]*;*lft2[+/-]* = 50/50, *lft1[+/-]*;*lft2[-/-]* = 43/43, *lft1[-/-]*;*lft2[+/-]* = 24/24 *lft1[-/-]*;*lft2[-/-]* = 0/55.

DOI: https://doi.org/10.7554/eLife.28785.003

The following figure supplements are available for figure 1:

**Figure supplement 1.** *lefty1[-/-]* mutants exhibit partially penetrant heart laterality defects.

DOI: https://doi.org/10.7554/eLife.28785.004

**Figure supplement 2.** *lefty* double mutants and morphants have distinct phenotypes.

DOI: https://doi.org/10.7554/eLife.28785.005

*1999*). Despite the ubiquity of this motif, the functions provided by coupling Nodal activation to inhibition remain unclear.

Several roles for Lefty feedback have been suggested. First, Nodal and Lefty were proposed to form a reaction-diffusion patterning system that regulates both mesendoderm formation and left/right patterning in zebrafish (*Chen and Schier, 2002*; *Kondo and Miura, 2010*; *Meinhardt, 2009*; *Müller et al., 2012*; *Schier, 2009*; *Shen, 2007*; *Shiratori and Hamada, 2006*). Notably, Nodal and Lefty fulfill the key biophysical requirements of a classical reaction-diffusion system: Both Nodal ligands (Cyclops and Squint) act as short-range (Cyclops) and mid-range (Squint) activators that induce their own expression as well as that of Lefty1 and Lefty2, which act as long-range, highly mobile inhibitors (*Chen and Schier, 2002*; *2001*; *Feldman et al., 2002*; *Meno et al., 1999*; *Müller et al., 2012*). Second, Lefty was argued to temporally restrict Nodal signaling by creating a 'window' of signaling competence (*van Boxtel et al., 2015*). In this model, Nodal signaling proceeds until sufficient Lefty accumulates to shut down further signaling. Third, theoretical studies suggest that inhibitory feedback has the potential to mitigate fluctuations in signaling (*Lander et al., 2009*). Deleterious increases or decreases in Nodal signaling could therefore be offset by adjustments in Lefty-mediated inhibition, ensuring robust development in the face of variation in the external environment or expression of pathway components.

To understand the role of inhibitory feedback in the Nodal/Lefty patterning system, we created embryos in which Nodal inhibition was decoupled from Nodal signaling. We found that inhibitory feedback mitigates signaling perturbations but is dispensable for development.

## Results

### Complete *lefty* loss causes lethal expansion of Nodal signaling and mesendoderm

To determine the consequences of removing feedback inhibition, we first used TALENs to generate null *lefty1* and *lefty2* alleles in zebrafish (*Figure 1C–T*) (*Bedell et al., 2012*; *Reyon et al., 2012*; *Sander et al., 2011*; *Sanjana et al., 2012*). *lefty1$^{a145}$* contains a 13-base-pair deletion that removes the translational start site and part of the predicted signal sequence (*Figure 1C*), and *lefty2$^{a146}$* contains an 11-base-pair deletion that results in a stop codon after amino acid 18 (*Figure 1D*). In contrast to wild type *lefty* mRNA, mutant *lefty* mRNAs were unable to induce Nodal loss-of-function phenotypes when injected into zebrafish embryos (*Figure 1E–J"*).

*lefty1$^{-/-}$* and *lefty2$^{-/-}$* single mutants were viable and exhibited no or only minor increases in mesendodermal gene expression (*Figure 1N,O,T*, *Figure 3—figure supplement 1E,F*), consistent with their overlapping early expression domains (*Figure 3—figure supplement 1A,B*). *lefty1$^{-/-}$* but not *lefty2$^{-/-}$* mutants had heart laterality defects (*Figure 1—figure supplement 1*), a hallmark of abnormal Nodal signaling (*Bakkers et al., 2009*) and reflecting their distinct spatial expression patterns during left-right patterning (*Bisgrove et al., 1999*).

A single functional *lefty* allele was sufficient for viability (*Figure 1Q,R,T*), but *lefty1$^{-/-}$;lefty2$^{-/-}$* double mutants exhibited severe patterning defects, including loss of heart, eyes, and tail (*Figure 1S*, *Figure 1—figure supplement 1A*). At the level of tissue patterning, *lefty1$^{-/-}$;lefty2$^{-/-}$* mutant embryos had expanded pSmad2 signaling gradients (*Figure 2*) prior to and during gastrulation: Signaling gradients had higher amplitudes and longer ranges in double mutants compared to wild type embryos (*Figure 2C*). Consistent with the expansion of the pSmad2 signaling gradient, *lefty1$^{-/-}$;lefty2$^{-/-}$* mutant embryos exhibited expanded expression of mesendodermal genes by gastrulation stages, whereas single mutants exhibited no or relatively minor upregulation (*Figure 3*, *Figure 3—figure supplement 1E,F*). Expanded mesendoderm has also been reported in *lefty* double morphants (*Agathon et al., 2001*; *Chen and Schier, 2002*; *Feldman et al., 2002*; *van Boxtel et al., 2015*), but *lefty1$^{-/-}$;lefty2$^{-/-}$* mutants differ in several aspects from morphants. For example, *lefty* morphants display disrupted gastrulation and do not survive past 24 hr, whereas *lefty* double mutants successfully gastrulate and survive past 24 hr (*Figure 1—figure supplement 2*). These differences are not caused by compensation in *lefty1$^{-/-}$;lefty2$^{-/-}$* mutants but by off-target morpholino effects: *lefty1$^{-/-}$;lefty2$^{-/-}$* mutants injected with *lefty1/2* morpholinos also display disrupted gastrulation and die by 24 hr (*Figure 1—figure supplement 2*). Together, our results demonstrate overlapping roles for *lefty1* and *lefty2* in restricting mesendoderm formation, but unique roles in left-right patterning.

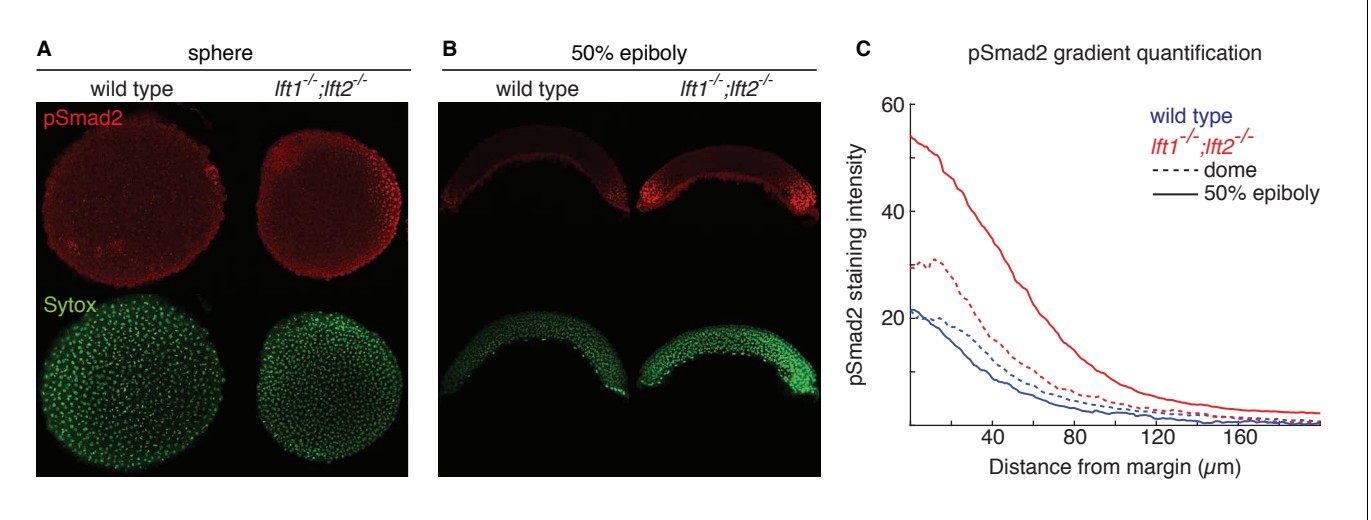

**Figure 2.** Lefty loss causes expanded Nodal signaling gradients. α-pSmad2 staining (red) at sphere stage (**A**, animal views) and 50% epiboly (**B**, lateral views) in representative wild type and *lft1⁻ᐟ⁻;lft2⁻ᐟ⁻* mutant embryos (nuclei labeled with Sytox nuclear stain (green)). Note stronger pSmad2 staining in *lft1⁻ᐟ⁻;lft2⁻ᐟ⁻* mutant embryos compared to wild type. (**C**) Quantification of pSmad2 gradients in wild type (blue) and *lft1⁻ᐟ⁻;lft2⁻ᐟ⁻* (red) mutant embryos at dome stage (dashed) and 50% epiboly (solid) shows an increase in the amplitude and range of the signaling gradient in *lft1⁻ᐟ⁻;lft2⁻ᐟ⁻* mutants. See Materials and methods and *Figure 2—figure supplement 1* for quantification details.

DOI: https://doi.org/10.7554/eLife.28785.006

The following figure supplement is available for figure 2:

**Figure supplement 1.** pSmad2 gradient quantification.

DOI: https://doi.org/10.7554/eLife.28785.007

## Mutations in *nodal* genes partially rescue *lefty1⁻ᐟ⁻;lefty2⁻ᐟ⁻* mutants

The patterning defects in *lefty* double mutants show a requirement for reduction of Nodal activity but do not establish a requirement specifically for inhibitory feedback. It is possible that a reduction in Nodal signaling by means other than inhibitory feedback could support patterning. For example, reducing *nodal* gene dosage could suppress *lefty* double mutant defects (*Chen and Schier, 2002*; *Feldman et al., 2002*; *Meno et al., 1999*). In support of this hypothesis, mutations in the Nodal genes *squint* or *cyclops* suppressed multiple aspects of the *lefty1⁻ᐟ⁻;lefty2⁻ᐟ⁻* mutant phenotype (*Figure 4*). *cyclops⁻ᐟ⁻;lefty1⁻ᐟ⁻;lefty2⁻ᐟ⁻* (*Figure 4C''*) and *squint⁻ᐟ⁻;lefty1⁻ᐟ⁻;lefty2⁻ᐟ⁻* (*Figure 4F''*) mutants formed eyes and full-length tails, structures missing in *lefty1⁻ᐟ⁻;lefty2⁻ᐟ⁻* mutants (*Figure 4A'', D'', M, N*). Moreover, upregulation of mesendodermal gene expression was suppressed in *squint⁻ᐟ⁻;lefty1⁻ᐟ⁻; lefty2⁻ᐟ⁻* mutants compared to *lefty1⁻ᐟ⁻;lefty2⁻ᐟ⁻* mutants (*Figure 4G–L'''*). Although these triple mutants are not viable, two functional *nodal* alleles are sufficient to generate mesendodermal gene expression patterns that are similar to those observed in wild type embryos with four *nodal* alleles and four *lefty* alleles. In previous studies, removal of *squint*, but not *cyclops*, partially suppressed defects in *lefty* double morphants (*Chen and Schier, 2002*; *Feldman et al., 2002*), but our results indicate that Lefty inhibits both Squint and Cyclops. The failure to fully rescue development may reflect an inability to precisely modulate Nodal dosage with this genetic approach, but reduction of Nodal dosage can partially rescue development in the complete absence of Lefty-mediated inhibition.

## Spatially decoupled ectopic *lefty* expression rescues *lefty1⁻ᐟ⁻;lefty2⁻ᐟ⁻* mutants

Nodal signaling induces expression and secretion of Lefty at the embryo margin (*Meno et al., 1999*) (*Figure 1A,B*, *Figure 3—figure supplement 1A,B*). To test the importance of the spatial coupling of *lefty* expression and Nodal signaling, we asked whether *lefty* needs to be induced where the Nodal pathway is active. We generated clones that expressed *lefty* or *lefty-gfp* ectopically in *lefty1⁻ᐟ⁻; lefty2⁻ᐟ⁻* mutant embryos, independent of Nodal signaling and outside of the endogenous *lefty*

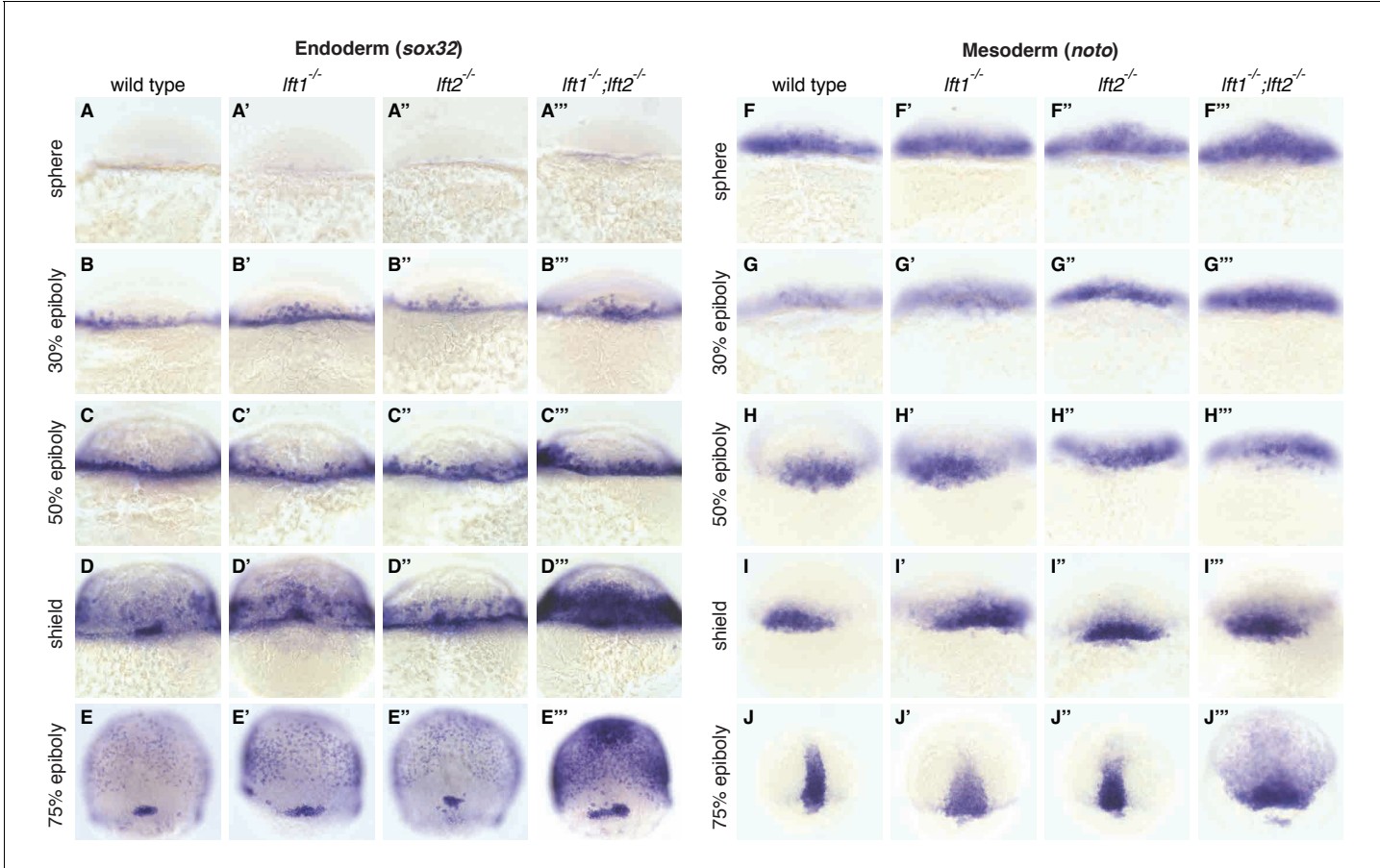

**Figure 3.** Lefty loss causes expanded expression of mesendodermal genes. *In situ* hybridization comparing expression of endodermal (*sox32/ casanova*) (A–E''') and mesodermal (*noto/floating head*) (F–J''') genes in wild type, *lft1⁻/⁻* mutants, *lft2⁻/⁻* mutants, and *lft1⁻/⁻;lft2⁻/⁻* mutants at the indicated developmental stages (dorsal views). Note upregulation of mesendoderm in *lft1⁻/⁻;lft2⁻/⁻* mutants starting at early gastrulation stages (e.g., shield stage). *lft1⁻/⁻;lft2⁻/⁻* embryos were generated from *lft1⁺/⁻;lft2⁻/⁻* incrosses, *lft1⁻/⁻;lft2⁻/⁻* incrosses, or *lft1⁻/⁻;lft2⁻/⁻* X *lft1⁺/⁻;lft2⁻/⁻* crosses. Embryos were genotyped after imaging (see Materials and methods for details).

DOI: https://doi.org/10.7554/eLife.28785.008

The following figure supplement is available for figure 3:

**Figure supplement 1.** Expression of *nodal*, *lefty*, and Nodal target genes in *lefty1⁻/⁻;lefty2⁻/⁻* mutants during gastrulation.

DOI: https://doi.org/10.7554/eLife.28785.009

expression domain (*Figure 5A,B*, *Figure 5—figure supplements 1* and *2*, *Figure 1B*, *Figure 3—figure supplement 1A,B*). We injected *lefty* mRNA into *lefty1⁻/⁻;lefty2⁻/⁻* mutant embryos and transplanted around 50 cells from these donor embryos into the animal pole of host *lefty1⁻/⁻;lefty2⁻/⁻* mutant embryos at sphere stage, when *lefty* expression normally commences (*Figure 5A,E,F*, *Figure 3—figure supplement 1A,B*). Strikingly, some of the *lefty1⁻/⁻;lefty2⁻/⁻* mutant hosts were rescued to normal morphology (*Figure 5C–F'*, *Figure 5—figure supplement 2*) and developed into fertile adults (see *Figure 5* legend for quantification). Thus, an ectopic, Nodal-independent source of Lefty at the animal pole can replace endogenous, Nodal-induced Lefty at the margin. The ability of ectopic Lefty-expressing clones to rescue *lefty1⁻/⁻;lefty2⁻/⁻* mutants is consistent with the high diffusivity and long-range, extracellular distribution of Lefty protein (*Chen and Schier, 2002*; *Marjoram and Wright, 2011*; *Müller et al., 2012*) (*Figure 1B*). In contrast to the requirement for spatially restricted Nodal signaling (*Figure 5—figure supplement 3*), the spatial coupling of *lefty* expression to Nodal signaling is not required for normal development.

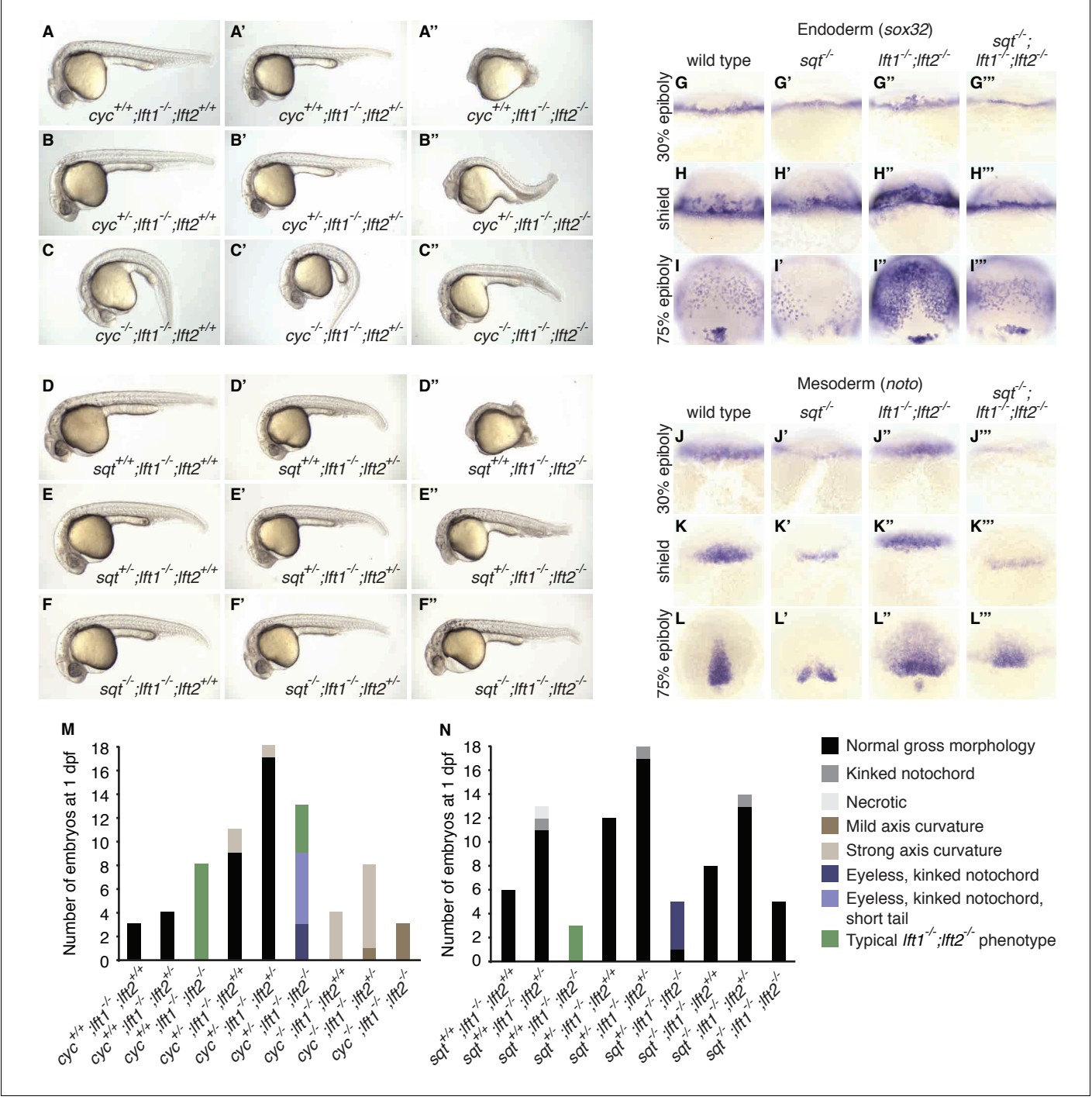

**Figure 4.** *nodal* mutations suppress *lefty1⁻/⁻;lefty2⁻/⁻* mutant defects. (A–F'') Lateral views of embryos with indicated genotypes at 1 day post-fertilization (dpf). Embryos are progeny from an incross of *cyc^{m294/+};lft1⁻/⁻;lft2⁺/⁻* (A–C'') or *sqt^{cz35/+};lft1⁻/⁻;lft2⁺/⁻* (D–F'') adults, and were genotyped after imaging (see Materials and methods for details). In contrast to *lft1⁻/⁻;lft2⁻/⁻* mutants (A'', D''), *nodal⁺/⁻;lft1⁻/⁻;lft2⁻/⁻* mutants (B'', E'') have long tails and well-defined heads, and some *nodal⁻/⁻;lft1⁻/⁻;lft2⁻/⁻* mutants (C'',F'') have eyes in addition to full-length tails. *nodal⁻/⁻;lft1⁻/⁻;lft2⁻/⁻* and *nodal⁺/⁻;lft1⁻/⁻;lft2⁻/⁻* mutants are not viable. *cyc* homozygotes exhibit the expected curved body axis (*Sampath et al., 1998*) (C,C'), but curvature is reduced in *cyc⁻/⁻;lft1⁻/⁻;lft2⁻/⁻* embryos (C''). (G–L''') *In situ* hybridization assessing expression of endodermal (*sox32/casanova*) (G–I''') or mesodermal (*noto/floating head*) (J–L''') genes in the indicated genotypes (dorsal views). Mesendoderm upregulation is less pronounced in *sqt⁻/⁻;lft1⁻/⁻;lft2⁻/⁻* compared to *lft1⁻/⁻;lft2⁻/⁻* mutants. (M–N) 72 embryos from a *cyc⁺/⁻;lft1⁻/⁻;lft2⁺/⁻* incross (M) and 84 embryos from a *sqt⁺/⁻;lft1⁻/⁻;lft2⁺/⁻* incross (N) were scored and imaged at 1 dpf, and subsequently genotyped. Number of embryos of each genotype with the indicated phenotype at 1 dpf is shown. Together, these results demonstrate that loss of *sqt* or *cyc* can suppress *lft1⁻/⁻;lft2⁻/⁻* mutant phenotypes.

DOI: https://doi.org/10.7554/eLife.28785.010

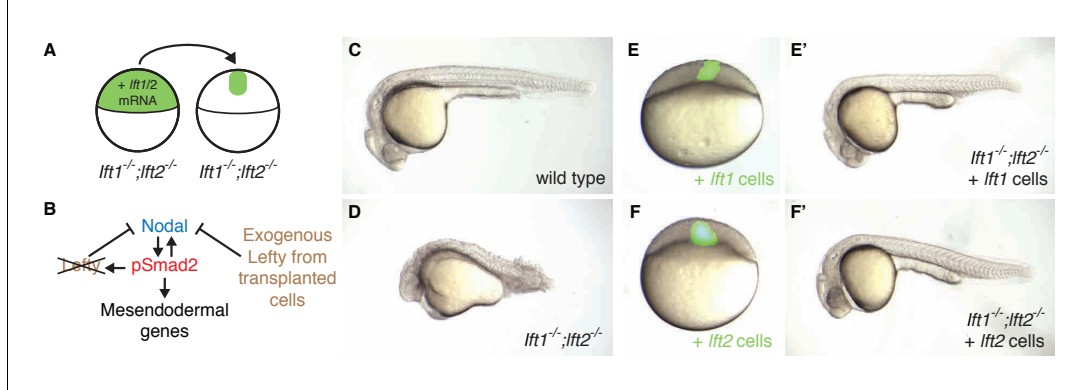

**Figure 5.** Ectopic *lefty* expression rescues Lefty loss. (A) *lft1⁻/⁻;lft2⁻/⁻* mutant donors were injected with 0.4 pg Alexa 488-dextran (3 kD, ThermoFisher) and 10 pg *lft1* or *lft2* mRNA at the one-cell stage. At sphere stage, around 50 cells from donors were transplanted into the animal pole of *lft1⁻/⁻;lft2⁻/⁻* mutant host embryos. (B) In host embryos Nodal is inhibited by exogenous, ectopic Lefty. (C,D) Untransplanted wild type (C) and *lft1⁻/⁻;lft2⁻/⁻* mutant (D) embryos at 1 day post-fertilization (dpf). (E–F') *lft1⁻/⁻;lft2⁻/⁻* mutant hosts 30–120 min after receiving cells from donors injected with Alexa 488-dextran + *lft1* (E) or *lft2* (F) mRNA and at 1 dpf (*lft1* mRNA (E'), *lft2* mRNA (F')). Note relatively normal morphology compared to untransplanted *lft1⁻/⁻;lft2⁻/⁻* sibling (D). Similar results were obtained using donors injected with *lft1-gfp* (200 pg) or *lft2-gfp* (60 pg) mRNA (data not shown). Number of embryos with indicated phenotype at 1 dpf/total number embryos receiving *lft1*-expressing cells: 21/45 normal gross morphology, 11/45 partially rescued (e.g. longer tail but no eyes), 6/45 cyclopia, 2/45 necrotic, and 5/45 dead. Number of embryos with indicated phenotype at 1 dpf/total number embryos receiving *lft2*-expressing cells: 19/45 normal gross morphology, 16/45 partially rescued, 2/45 cyclopia, 3/45 typical *lft1⁻/⁻;lft2⁻/⁻* mutant phenotype, 1/45 necrotic, and 4/45 dead.

DOI: https://doi.org/10.7554/eLife.28785.011

The following figure supplements are available for figure 5:

**Figure supplement 1.** Transplanted cells are mostly retained in the animal pole during germ layer patterning.
DOI: https://doi.org/10.7554/eLife.28785.012

**Figure supplement 2.** Transplanted cells frequently populate head structures.
DOI: https://doi.org/10.7554/eLife.28785.013

**Figure supplement 3.** Ubiquitous Activin expression can partially rescue maternal-zygotic *one-eyed pinhead* embryos.
DOI: https://doi.org/10.7554/eLife.28785.014

## Spatiotemporally decoupled Nodal inhibition rescues *lefty1⁻/⁻; lefty2⁻/⁻* mutants

Induction of *lefty* expression by Nodal signaling couples pathway activation to inhibition. To test whether patterning can occur when Nodal pathway inhibition is spatially *and* temporally decoupled from Nodal activity, we attempted to replace the inhibitory activity of Lefty with a small molecule drug, SB-505124, that selectively inhibits Nodal signaling by preventing ATP from binding to Nodal receptors (*Figure 6A*) (*DaCosta Byfield et al., 2004*; *Fan et al., 2007*; *Hagos and Dougan, 2007*; *Hagos et al., 2007*; *van Boxtel et al., 2015*; *Vogt et al., 2011*). We exposed *lefty1⁻/⁻;lefty2⁻/⁻* mutants to low concentrations of the Nodal inhibitor drug starting at three developmental stages: (1) the 8 cell stage, 3 hr before the onset of *nodal* and *lefty* expression, (2) sphere stage, when *nodal* and *lefty* expression normally begins, and (3) shield stage, 2.5 hr after expression has commenced. Strikingly, exposure of *lefty1⁻/⁻;lefty2⁻/⁻* mutants to inhibitor drug resulted in phenotypically normal embryos that developed into fertile adults (*Figure 6B–I*, *Figure 6—figure supplements 1* and *2*). Different drug concentrations were required for rescue depending on the timing of treatment: Mutants exposed at earlier times were rescued by lower concentrations of Nodal inhibitor drug than mutants exposed at later times (*Figure 6I*, *Figure 6—figure supplements 1* and *2*). Notably, even drug treatment starting hours before or after the onset of *nodal* and *lefty* expression rescued *lefty1⁻/⁻; lefty2⁻/⁻* mutants (*Figure 6D,F,I*, *Figure 6—figure supplements 1* and *2*). These results show that highly specific timing or progressively increasing levels of inhibition during embryogenesis are not required for mesendoderm development.

To determine the mechanism by which Nodal inhibitor-treated mutants are rescued, we analyzed Smad2 phosphorylation and mesendodermal gene expression. Interestingly, despite their eventual rescue, *lefty1⁻/⁻;lefty2⁻/⁻* mutants that were exposed to inhibitor before the onset of Nodal signaling

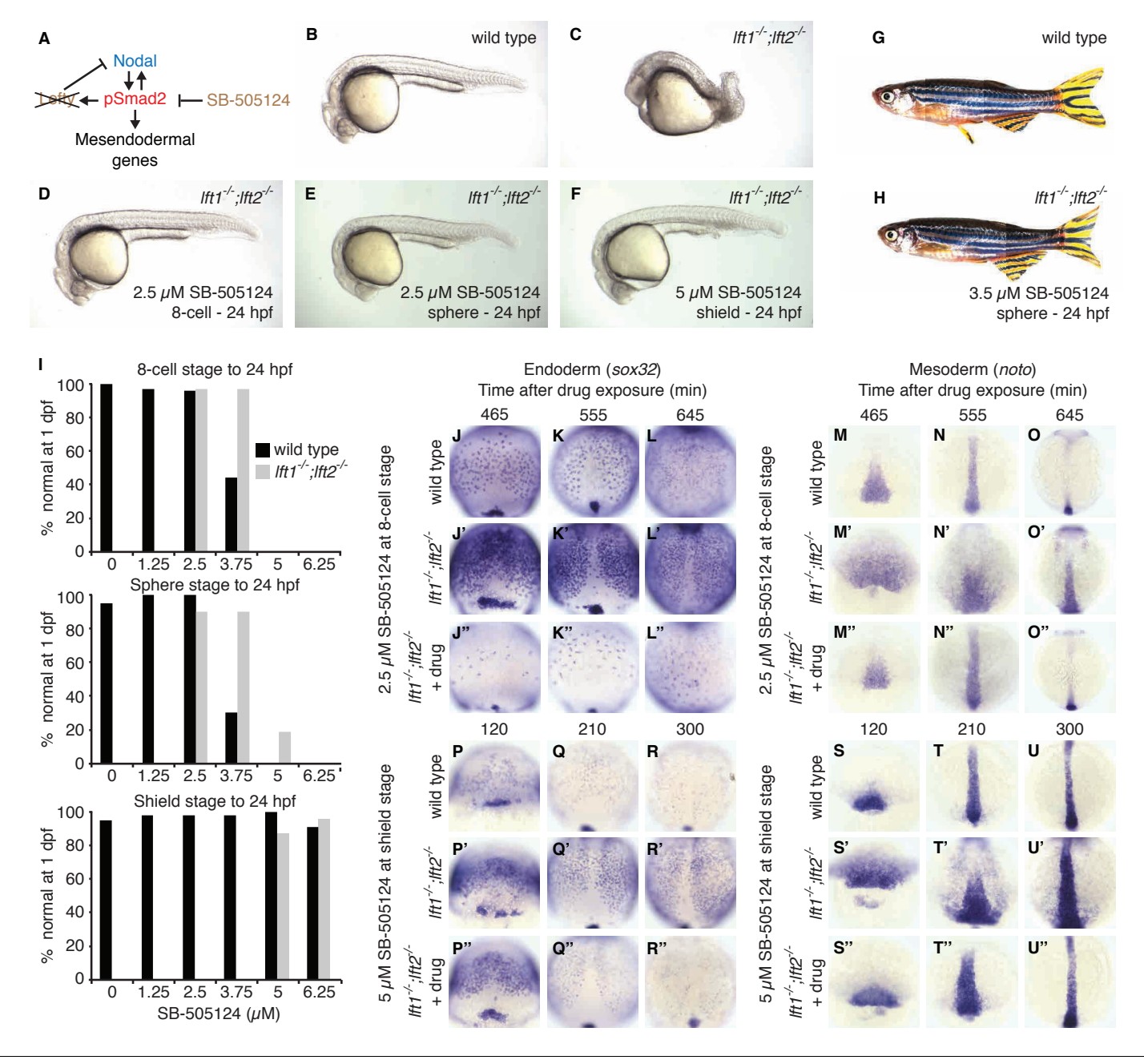

**Figure 6.** Artificial, uniform Nodal inhibition rescues Lefty loss. Wild type and *lft1⁻/⁻;lft2⁻/⁻* mutant embryos were exposed to 0, 1.25, 2.5, 3.75, 5, or 6.25 µM SB-505124 (Nodal inhibitor drug) starting at the 8 cell stage, sphere stage, or shield stage. (Exposure to 30–50 µM fully inhibits Nodal signaling (*Fan et al., 2007*; *Hagos et al., 2007*; *Hagos and Dougan, 2007*).) Drug was removed at 24 hr post-fertilization (hpf). (A) Inhibition is decoupled from signaling in drug-treated *lft1⁻/⁻;lft2⁻/⁻* mutants. (B–F) Exposure to 2.5 µM SB-505124 at the 8 cell (D) or sphere (E) stage or 5 µM at shield stage (F) rescues gross morphological defects in *lft1⁻/⁻;lft2⁻/⁻* mutants (C); compare to wild type embryo (B). Embryos were imaged at 1 day post-fertilization (dpf). (G,H) Drug exposure rescues *lft1⁻/⁻;lft2⁻/⁻* mutants to adulthood. (G) 10 month old wild type TE adult male. (H) 5 month old *lft1⁻/⁻;lft2⁻/⁻* mutant male treated with 3.5 µM SB-505124 from sphere stage to 24 hpf. (I) Percent of wild type (black bars) or double mutant (gray bars) embryos with normal gross morphology at 1 dpf after drug exposure is plotted. See *Figure 6—figure supplements 1* and *2* for further details. (J–U'') Endodermal (*sox32/casanova*) and mesodermal (*noto/floating head*) gene expression in embryos treated as indicated was assessed using *in situ* hybridization. *lft1⁻/⁻;lft2⁻/⁻* mutants treated starting at the 8 cell stage (2.5 µM) exhibit decreases in endoderm compared to wild type embryos (J–L''), but have mesodermal gene expression similar to wild type embryos (M–O''). *lft1⁻/⁻;lft2⁻/⁻* mutants treated at shield stage (5 µM) exhibit upregulation of mesendoderm until ~5 hr post-exposure (P–U''). See *Figure 6—figure supplement 3* for *in situ* hybridization of embryos treated with drug at sphere and shield stage, as well as additional time points and treatments.

*Figure 6 continued on next page*

*Figure 6 continued*

DOI: https://doi.org/10.7554/eLife.28785.015

The following figure supplements are available for figure 6:

**Figure supplement 1.** Artificial, uniform Nodal inhibition rescues morphology at 1 dpf in *lefty1⁻/⁻;lefty2⁻/⁻* mutants.

DOI: https://doi.org/10.7554/eLife.28785.016

**Figure supplement 2.** Artificial, uniform Nodal inhibition rescues morphology at 4 dpf in *lefty1⁻/⁻;lefty2⁻/⁻* mutants.

DOI: https://doi.org/10.7554/eLife.28785.017

**Figure supplement 3.** Artificial, uniform Nodal inhibition rescues mesendodermal gene expression in *lefty1⁻/⁻;lefty2⁻/⁻* mutants.

DOI: https://doi.org/10.7554/eLife.28785.018

**Figure supplement 4.** Nodal activity gradients in under- and over-rescued *lefty* double mutants.

DOI: https://doi.org/10.7554/eLife.28785.019

not only showed reduced Smad2 phosphorylation compared to double mutants, but initially displayed reduced Smad2 phosphorylation and endodermal gene expression compared to wild type (*Figure 6J–O''*, *Figure 2—figure supplement 1*, *Figure 6—figure supplement 3A,B*). The mechanism by which premature inhibitor exposure rescues development therefore involves a delay in full activation of Nodal signaling. Highlighting the sensitivity to Nodal inhibitor drug concentration, *lefty* double mutants exposed to excess or sub-rescuing doses exhibited diminished or expanded pSmad2 activity gradients, respectively, with corresponding Nodal loss- or gain-of-function phenotypes (*Figure 6—figure supplements 1* and *4*). Mutants exposed to inhibitor drug after the onset of Nodal signaling exhibited excess mesendodermal gene expression until ~5 hr post-exposure (*Figure 6P–U''*, *Figure 6—figure supplement 3E,F*), but they ultimately developed normally (*Figure 6F,I*, *Figure 6—figure supplements 1* and *2*). Together, these results demonstrate that the precise spatial and temporal coupling of *lefty* expression and Nodal activity is not essential for normal development.

## Inhibitory feedback can mitigate signaling perturbations

The rescue of *lefty* mutants with a Nodal inhibitor drug shows that inhibitory feedback is not a requirement for mesendoderm patterning. This result leaves open the possibility that inhibitory feedback enhances developmental robustness by mitigating signaling fluctuations (*Lander et al., 2009*). Perturbations that decrease Nodal signaling might be corrected by a compensatory decrease in activation of Lefty, thus restoring the activator/inhibitor balance. To test this model, we challenged wild type embryos with exogenous Nodal pathway activation or inhibition. Injecting low levels of *lefty1* mRNA into wild type embryos dramatically reduced *lefty2* transcript abundance (*Figure 7A,C,B,D*), but left expression of the mesoderm marker *noto* relatively intact (*Figure 7A', C', B', D'*). Conversely, increasing signaling by injecting mRNA encoding constitutively-active *smad2* (*CA-smad2*, (*Baker and Harland, 1996*; *Dick et al., 2000*; *Gritsman et al., 1999*; *Müller et al., 1999*)) resulted in a marked increase in *lefty2* expression (*Figure 7E,G,F,H*), but unchanged *noto* expression (*Figure 7E', G', F' and H'*). Wild type embryos thus appear to compensate for perturbed Nodal signaling by sensitively adjusting *lefty* levels.

This model makes a key prediction: If Lefty feedback allows the embryo to correct signaling perturbations, embryos with compromised feedback should be more sensitive to Nodal signaling challenges. To test this hypothesis, we assessed developmental outcomes after manipulating Nodal signaling levels in wild type embryos and in drug-treated *lefty1⁻/⁻;lefty2⁻/⁻* mutants, in which inhibition is decoupled from signaling (*Figure 6A*). Wild type embryos challenged with a small dose of *lefty1* mRNA exhibited normal phenotypes at 24 hpf (*Figure 8A,B*) as well as normal mesendodermal gene expression at 50% epiboly and shield stages (*Figure 8H–I'*). In contrast, challenging feedback-compromised embryos with the same dose of *lefty1* mRNA led to markedly decreased *noto* expression (*Figure 8I,K,M,I', K', M'*), and 24 hpf phenotypes resembling partial loss-of-function Nodal mutants (*Figure 8D,E*) (*Feldman et al., 1998*; *Gritsman et al., 1999*). Challenging with a modest increase in Nodal signaling revealed a similar difference in sensitivity. Wild type embryos tolerated a small dose of *CA-smad2* mRNA (*Figure 8C*), while drug-rescued *lefty* double mutants developed severe phenotypic defects in response to the same treatment (*Figure 8F*), although changes in early

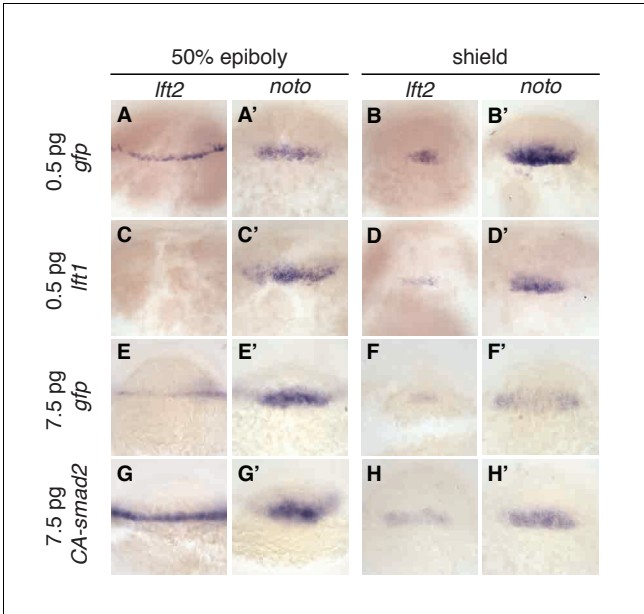

**Figure 7.** Expression of *lefty2* responds sensitively to Nodal signaling perturbations. Wild type embryos at the one-cell stage were injected with 0.5 pg *gfp* mRNA (first row, (**A–B'**)), 0.5 pg *lft1* mRNA (second row, (**C–D'**)), 7.5 pg *gfp* mRNA (third row, (**E–F'**)), or 7.5 pg *CA-smad2* mRNA (fourth row, (**G–H'**)). Embryos were fixed at 50% epiboly and shield stages and expression of endogenous *lft2* (**A,B,C,D,E,F,G,H**) and the mesoderm marker *noto* (**A',B'C',D',E',F',G'H'**) were assessed using *in situ* hybridization. Note the dramatic decrease and increase in *lft2* expression after *lft1* (**A,B,C,D**) and *CA-smad2* injection (**E,G,F,H**), respectively. Expression of *noto* was relatively unchanged in all treatments.

DOI: https://doi.org/10.7554/eLife.28785.020

mesendodermal gene expression were relatively minor (*Figure 8N–S'*). These results reveal that Lefty-mediated inhibitory feedback can mitigate aberrant fluctuations in Nodal signaling levels (*Figure 8G*).

## Discussion

The results in this study show that inhibitory feedback in the Nodal/Lefty system stabilizes Nodal signaling but is not essential for mesendoderm patterning and viability. The rescue of *lefty* mutants by ectopic *lefty* expression (*Figure 5*, *Figure 5—figure supplements 1* and *2*) and exposure to Nodal inhibitor drug (*Figure 6*, *Figure 6—figure supplements 1–4*, *Figure 2—figure supplement 1*) is consistent with the high diffusivity of Lefty measured previously (*Müller et al., 2012*), but is surprising in light of the functions assigned to inhibitory feedback. Specifically, inhibitory feedback has been implicated in (1) shutting down pathway activity at the appropriate time to generate a pulse or window of signaling, (2) shaping spatial signaling profiles, (3) acting as part of self-organizing reaction-diffusion systems, and (4) mitigating fluctuations in signaling activity. Below, we discuss our results in the context of these models.

First, inhibitory feedback has been suggested to turn off signaling activity when it is no longer needed (*Dessaud et al., 2007*; *Freeman, 2000*; *Golembo et al., 1996*; *Ribes and Briscoe, 2009*; *Shiratori and Hamada, 2006*). For Nodal-mediated patterning, it has been proposed that progressively increasing Lefty levels shut down Nodal signaling at the onset of gastrulation (*van Boxtel et al., 2015*). However, we find that Nodal signaling is already increased in *lefty* double mutants by sphere stage, suggesting an earlier role for Lefty (*Figure 2*). Moreover, Lefty can be replaced by an inhibitor drug added as early as the 8 cell stage (*Figure 6*, *Figure 6—figure supplements 1–4*, *Figure 2—figure supplement 1*), demonstrating that mesendoderm patterning can proceed without progressively increasing inhibition and without temporally precise feedback inhibition. Our results

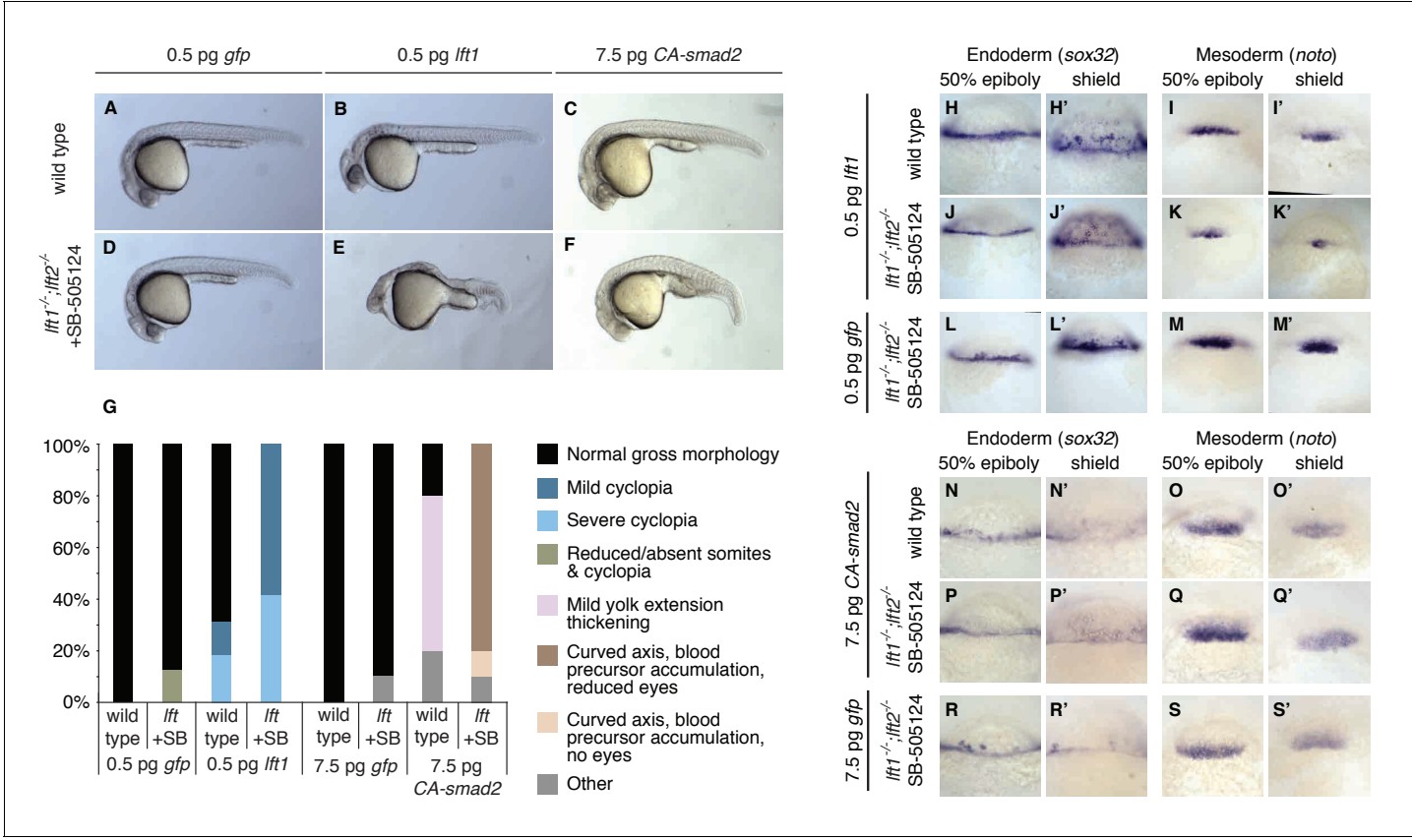

**Figure 8.** Correction of Nodal signaling challenges requires Lefty feedback. (**A–F**) Wild type embryos with intact feedback circuits (**A–C**) and feedback-decoupled drug-treated *lft1*^-/-^;*lft2*^-/-^ mutants (**D–F**) were challenged by decreasing Nodal signaling with injection of 0.5 pg *lft1* mRNA (**B,E**) or by increasing Nodal signaling with injection of 7.5 pg *CA-smad2* (**C,F**). *lft1*^-/-^;*lft2*^-/-^ mutant embryos were exposed to 2 µM SB-505124 (Nodal inhibitor drug) at the 8 cell stage and phenotypes were assessed at 1 dpf. 0.5 pg *lft1* causes mild Nodal loss-of-function phenotypes in wild type embryos (**B**), but strong Nodal loss-of-function phenotypes such as cyclopia and loss of head and trunk mesendoderm in feedback-decoupled embryos (**E**). *CA-smad2* is well-tolerated in wild type embryos (**C**), but causes reduction of eyes, axis curvature, and accumulation of blood precursors in feedback-decoupled embryos (**F**). (**G**) Percentages of embryos with indicated phenotypes. Total embryos analyzed, *lft* challenge experiment: wild type + 0.5 pg *gfp* mRNA = 10, wild type +0.5 pg *lft1* mRNA = 16, drug-treated *lft1*^-/-^;*lft2*^-/-^ mutants + 0.5 pg *gfp* mRNA = 16, drug-treated *lft1*^-/-^;*lft2*^-/-^ mutants + 0.5 pg *lft1* mRNA = 12. Total embryos analyzed, *CA-smad2* challenge experiment: wild type +7.5 pg *gfp* mRNA = 18, wild type +7.5 pg *CA-smad2* mRNA = 20, drug-treated *lft1*^-/-^;*lft2*^-/-^ mutants + 7.5 pg *gfp* mRNA = 19, drug-treated *lft1*^-/-^;*lft2*^-/-^ mutants + 7.5 pg *CA-smad2* mRNA = 20. (**H–M'**) Wild type (**H–I'**) and *lft1*^-/-^;*lft2*^-/-^ mutant embryos (**J–M'**) were injected with 0.5 pg *gfp* or *lft1* mRNA at the one-cell stage. Mutants were exposed to 2 µM Nodal inhibitor drug SB-505124 starting at the 8 cell stage, and expression of the endoderm marker *sox32* (**H, H', J, J', L, L'**) and the mesoderm marker *noto* (**I, I', K, K', M, M'**) were assessed at 50% epiboly and shield stage as indicated via *in situ* hybridization. Note the decreased expression of *noto* in drug-treated double mutants injected with *lft1* mRNA (**K, K'**) compared to wild type embryos injected with *lft1* mRNA (**I,I'**). (**N–S'**) Wild type (**N–O'**) and *lft1*^-/-^;*lft2*^-/-^ mutant embryos (**P–S'**) were injected with 7.5 pg *gfp* or *CA-smad2* mRNA at the one-cell stage. Mutants were exposed to 2 µM Nodal inhibitor drug SB-505124 starting at the 8 cell stage, and expression of *sox32* (**N,N', P, P', R, R'**) and *noto* (**O, O', Q, Q',S, S'**) were assessed at 50% epiboly and shield stage as indicated via *in situ* hybridization.

DOI: https://doi.org/10.7554/eLife.28785.021

The following figure supplement is available for figure 8:

**Figure supplement 1.** Response of wild type and *lefty1*^-/-^;*lefty2*^-/-^ mutant embryos to *squint* injection.

DOI: https://doi.org/10.7554/eLife.28785.022

do not rule out that Lefty accumulation shuts down Nodal signaling during normal development, but they do indicate that this is not an absolute requirement for patterning.

Second, inhibitory feedback has been implicated in shaping the spatial profile of pathway activity (*Barkai and Shilo, 2009*; *Ben-Zvi et al., 2008*; *Freeman, 2000*; *Gerlitz and Basler, 2002*; *Lecuit and Cohen, 1998*; *Piddini and Vincent, 2009*; *Ribes and Briscoe, 2009*; *Schilling et al., 2012*; *van Boxtel et al., 2015*). In drug-rescued *lefty1*^-/-^;*lefty2*^-/-^ mutants, Nodal activity gradients and mesendodermal gene expression patterns initially differed from wild type (*Figure 6J–U''*,

*Figure 2—figure supplement 1*, *Figure 6—figure supplements 3* and *4*). However, rescued mutants developed into fertile adults (*Figure 6H*, *Figure 6—figure supplement 2*), demonstrating that precise wild type activity gradients and mesendodermal gene expression patterns during early embryogenesis are not essential for germ layer development. Inhibitory feedback may subtly shape gene expression patterns, but the patterns achieved without feedback are a suitable starting point for successful development.

Third, the dispensability of inhibitory feedback in the Nodal/Lefty system raises questions about the role of Nodal and Lefty as an activator/inhibitor pair in self-organizing reaction-diffusion models. Although Nodal and Lefty fulfill the key regulatory and biophysical requirements of a short-to-mid-range autocatalytic activator and a long-range feedback inhibitor (*Hamada, 2012*; *Meno et al., 1999*; *Müller et al., 2012*; *Schier, 2009*), our finding that development can be normal without inhibitory feedback indicates that this system does not require the ability to form self-organizing reaction-diffusion patterns. Instead, it is conceivable that the pre-patterning of the early embryo by maternal factors and the local activation of Nodal eliminate the need for self-organizing pattern generation by the Nodal/Lefty circuit. In this scenario, Nodal and Lefty may have constituted a reaction-diffusion activator/inhibitor pair in ancestral organisms but, through the addition of other regulatory layers, mesendoderm patterning lost the requirement for inhibitory feedback.

Finally, feedback inhibition has been implicated in buffering fluctuations in pathway activity (*Barkai and Shilo, 2009*; *Eldar et al., 2003*; *Lander et al., 2009*). Feedback may be required to optimize inhibitor levels, as suggested by the narrow range (~2 fold) of inhibitor concentrations that rescue Lefty loss (*Figure 6I*, *Figure 6—figure supplements 1* and *2*). Indeed, the adjustment of *lefty* expression in response to slight alterations in Nodal signaling (*Figure 7*) and the failure of feedback-decoupled embryos to cope with perturbations in Nodal signaling (*Figure 8*) support this idea. The ability to dynamically adjust pathway activity may allow the embryo to create reliable patterns in the face of endogenous signaling fluctuations and uncertain environmental conditions. We note, however, that Lefty feedback does not protect the embryo against all perturbations: Drug-rescued *lefty* mutants actually fared better than wild type embryos when challenged with injection of *squint* mRNA (*Figure 8—figure supplement 1*). Dissecting why Lefty feedback corrects some perturbations but not others will provide a window into the mechanisms and limits of robust patterning.

Our results have implications not only for the roles of feedback inhibition during development, but also demonstrate the feasibility of preventing patterning defects with small molecule drug exposure. Although suggested applications to human embryos might currently seem fanciful and would be challenging and fraught with ethical concerns, embryos bearing compromised patterning circuits could be identified by sequencing a single embryonic cell, and birth defects could be prevented by exposure to the appropriate small molecule. More generally, our study adds a new facet to recent revisions of classical patterning models (*Alexandre et al., 2014*; *Chen et al., 2012*; *Dominici et al., 2017*; *Dubrulle et al., 2015*; *Varadarajan et al., 2017*). For example, a tethered form of Wingless can replace endogenous Wingless, challenging models in which a gradient of diffusing Wingless is indispensable for tissue patterning (*Alexandre et al., 2014*). In the same vein, our observations challenge models in which inhibitory feedback is an absolute requirement for patterning and viability, but support the idea that inhibitory feedback enhances robustness by stabilizing signaling during development.

## Materials and methods

### Generation of *lefty* mutants

Mutations in *lefty* genes were induced using TALENs (*Bedell et al., 2012*; *Sander et al., 2011*; *Sanjana et al., 2012*). The *lefty1* TALEN pair was generated using the FLASH assembly kit (*Reyon et al., 2012*); the *lefty2* TALEN pair was generated using the TALE Toolbox (*Sanjana et al., 2012*).

 *lefty* TALENs target sites:
 *lefty1* TALEN L: tcctgcaccttgaaaaga
 *lefty1* TALEN R: tgcgcaaaggaggcacgc
 *lefty2* TALEN L: ttcatccagctgttcatttt
 *lefty2* TALEN R: tgctggaatccctgtgtgag

Embryos from an incross of the TLAB wild type strain were injected at the one-cell stage with 300–450 pg mRNA encoding each TALEN pair. Injected fish were grown using standard fish husbandry protocols and fin clipped as adults. Genomic DNA was generated from fin material using the Hot Shot method (*Meeker et al., 2007*). To identify animals carrying mutations, PCR using primers flanking the target sites was carried out and the resulting amplicons were re-annealed and digested with mismatch-cleaving T7 endonuclease I (NEB) (*Mussolino et al., 2011*). PCR products from positive animals were cloned using a TOPO TA kit (Life Technologies) and sequenced. Positive animals were outcrossed and progeny were sequenced and tested for germline transmission.

Primers flanking lefty TALEN target sites:

*lefty1* forward primer: catgtatcaccttccctctgatgtc

*lefty1* reverse primer: gcattagcctatatgttaacttgcac

*lefty2* forward primer: tacttatcaacatgagcatcaatgg

*lefty2* reverse primer: gaattgtgcataagtaacccacctg

## Genotyping

Genomic DNA was generated using the Hot Shot method (*Meeker et al., 2007*).

*lefty1*: The 13-base-pair deletion in *lefty1*[a145] destroys a PshAI restriction site. To genotype the *lefty1* locus, PCR amplicons were generated using primers flanking the deletion and subsequently digested with PshAI endonuclease (NEB). Genotyping primers were identical to the *lefty1* forward/reverse primers described above. Complete digestion by PshAI indicates that both alleles are wild type, partial digestion indicates heterozygosity, and failure to digest indicates homozygosity for the *lefty1*[a145] mutation.

*lefty2:* The 11-base-pair deletion in *lefty2*[a146] was detected using a mutant-specific forward primer that spans the deletion. A forward primer specific to the wild type allele was also designed, as well as a reverse primer that is fully complementary to both alleles. To genotype the *lefty2* locus, PCR was carried out using either the wild type- or mutant-specific forward primer and the common reverse primer. A band with the wild type- but not mutant-specific primer indicates that both alleles are wild type, bands with both primer sets indicate heterozygosity, and a band with the mutant- but not wild type-specific primers indicates homozygosity for the *lefty2*[a146] mutation. Optimal PCR conditions: Taq polymerase, 25 cycles, 57°C annealing temperature.

*lefty2* wild type genotyping forward primer: cattttgaccacagcgat

*lefty2* mutant genotyping forward primer: gttcattttgaccactcac

The common reverse primer was identical to *lefty2* reverse primer described above.

*squint:* The *squint*[cz35] allele has a ~ 1.9 kb insertion in exon 1, and was detected as in (*Feldman et al., 1998*).

*cyclops*: The *cyclops*[m294] mutation destroys an AgeI restriction site, and was detected as in (*Sampath et al., 1998*).

## *lefty* and *squint* expression constructs

To generate mRNA from all constructs, plasmids were linearized with NotI-HF endonuclease (NEB) and purified using a Qiagen PCR clean-up kit. Capped mRNA was generated from linearized plasmid using an SP6 mMessage mMachine kit (Ambion) and purified with a Qiagen RNeasy kit.

After purification, mRNA was quantified using a NanoDrop spectrophotometer (ThermoFisher) and diluted to the appropriate concentration. For microinjections, a micrometer was used to adjust the drop volume to 0.5 nl. Depending on the concentration of the injection mix, a total volume of 1–2 nl was injected per embryo.

Lefty1-GFP, Lefty2-GFP, and untagged Lefty1 and Lefty2 constructs used in *Figures 5* and *7* were identical to those in (*Müller et al., 2012*). These constructs lack endogenous UTRs and contain the consensus Kozak sequence gccacc immediately preceding the start codon.

Allele activity experiments: To determine whether mutant *lefty* alleles retain Nodal inhibitory activity, wild type and mutant *lefty* mRNA was injected into wild type embryos and Nodal loss-of-function phenotypes were assessed (*Figure 1*).

The 13 bp *lefty1*[a145] mutation removes part of the endogenous Kozak sequence (gaaaag). Therefore, *lefty1* constructs containing this endogenous Kozak were generated, rather than the consensus Kozak sequence gccacc as in (*Müller et al., 2012*). Primers with either the endogenous Kozak

sequence (for the wild type construct) or the truncated endogenous sequence and deleted region of coding sequence (for the mutant construct) were designed, and the Lefty1 construct from (*Müller et al., 2012*) was used as a PCR template. The resulting fragments were cloned into BamHI and XhoI sites in pCS2(+). Both constructs lack endogenous UTRs.

The *lefty2* wild type construct was the same used in (*Müller et al., 2012*) and in the transplantation experiments in *Figure 5*. The *lefty2^a146* construct was made by generating cDNA from *lefty2* homozygous embryos, amplifying the mutant *lefty2* coding sequence, and cloning the resulting fragment into ClaI and XhoI in the pCS2(+) vector. In addition to the 11 bp deletion, the mutant construct contains three silent SNPs at position 184 (T->C), 932 (A->C), and 943 (T->A). Both constructs lack endogenous UTRs and contain the consensus Kozak sequence gccacc immediately preceding the start codon.

Nodal overexpression experiment: The construct used to generate *squint* mRNA in the Nodal overexpression experiment (*Figure 8—figure supplement 1*) was identical to that used in (*Müller et al., 2012*). This construct lacks endogenous UTRs and contains the consensus Kozak sequence gccacc immediately preceding the start codon.

## α-pSmad2 immunofluorescence

The protocol was modified from (*van Boxtel et al., 2015*). Briefly, embryos were fixed in 4% formaldehyde (in 1x PBS) overnight at 4°C, washed in PBST (1x PBS + 0.1% (w/v) Tween 20), manually deyolked, dehydrated in a MeOH/PBST series (25%, 50%, 75%, and 100% MeOH), and stored at −20°C until staining. To prepare for staining, embryos were rehydrated in a MeOH/PBSTr (1x PBS + 1% (w/v) Triton X-100) series (75%, 50%, and 25% MeOH), washed 3x in PBSTr, and incubated for 20 min in ice-cold acetone. Embryos were then washed 3x in PBSTr, incubated in antibody binding buffer (PBSTr +1% (v/v) DMSO) for two hours at room temperature, then incubated overnight at 4°C with a 1:1000 dilution of α-pSmad2 antibody (Cell Signaling Technology #8828, Danvers, MA, USA) in antibody binding buffer. After primary treatment, embryos were washed 6x in PBSTr, incubated in antibody binding buffer for 30 min at room temperature, and incubated for two hours at room temperature with a 1:2000 dilution of goat α-rabbit Alexa 647 conjugate (ThermoFisher A-21245) in PBSTr +1% (v/v) DMSO. Embryos were then washed 6x in PBSTr, 3x in PBS and incubated with 200 nM Sytox green in PBS for 30 min at room temperature. Finally, embryos were washed 3x in PBS and dehydrated in a MeOH/PBS series (50% and 100% MeOH). Stained embryos were stored at −20°C in 100% MeOH until imaging.

## α-pSmad2 imaging

Embryos were mounted in agarose and cleared with 2:1 benzyl benzoate:benzyl alcohol (BBBA) (*Yokomizo et al., 2012*). Briefly, a dehydrated embryo was dropped into molten low-melting point agarose (1% (w/v) in H$_2$O), transferred onto a coverglass and oriented manually. All embryos (other than sphere stage) were mounted 'margin down' (i.e. with the animal/vegetal axis parallel to the coverglass, see below for rationale). Shield stage embryos were rolled to ensure that the dorsal/ventral axis was parallel to the coverglass. Sphere stage embryos were mounted with the animal/vegetal axis perpendicular to the coverglass (animal pole facing up). The agarose drop was then dehydrated with three washes of 100% MeOH, two washes of 50:50 (v/v) BBBA:MeOH, and 3 washes of BBBA. Cleared embryos were then sealed to a microscope slide using fast wells reagent reservoirs (Grace Bio-Labs). Imaging was performed on Sytox Green and Alexa 647 channels using an LSM 700 confocal microscope (20x air objective, 0.5 NA). Image stacks extended from the embryo margin (adjacent to the coverglass) to beyond the center of the embryo. Z-planes were spaced at 2 μm intervals.

## α-pSmad2 image segmentation and quantification

Quantification of pSmad2 and Sytox green staining intensity in laterally-mounted embryos was performed using the ten z-slices surrounding the center of the embryo axis (i.e. five slices above and five slices below the embryo center). This region was chosen to minimize artifacts due to light scattering, which causes a decrease in apparent fluorescence intensity in deeper tissue planes. Our procedure—looking only at slices close to the 'central plane' of the embryo—allows the entire gradient to be sampled within each slice, and ensured that all data were taken from planes within a narrow range of imaging depths, effectively controlling for signal drop-off with imaging depth.

Nuclei were segmented from the Sytox Green channel images using a custom pipeline implemented in MATLAB (*Source code 1*). Briefly, out-of-plane background signal was approximated by blurring adjacent z-slices (i.e. the slice above and below the plane being segmented) with a Gaussian smoothing kernel and summing. This background was subtracted from the segmentation image, and preliminary boundaries for nuclei were identified by adaptive thresholding (http://homepages.inf.ed.ac.uk/rbf/HIPR2/adpthrsh.htm) of the resulting image. Spurious objects were discarded by morphological filtering (based on object size). Final segmentation boundaries were defined after manual checking and correction with a custom MATLAB script (*Source code 1*). The fluorescence intensity of each segmented nucleus was defined as the mean intensity of its constituent pixels.

The distance of each nucleus from the margin was defined along a curved embryo contour (*Figure 2—figure supplement 1*). This contour was defined by 1) projecting all segmented nuclei centroids onto a single z-plane, 2) creating a full embryo 'mask' by filling a convex hull containing all of these points, 3) identifying the left and right margin boundaries as the points of maximum curvature on the convex hull, 4) taking the distance transform of the embryo mask, and 5) stepping along the 'valley' of the distance transform that connects the left and right margins (as defined above). This rough contour was then smoothed using a Savitsky-Golay filter to yield the final contour. The position of each nucleus was then projected onto the contour, and the distance from the margin (as plotted in *Figure 2C* and *Figure 2—figure supplement 1*) was determined as the distance to the closest margin along this curve.

### *In situ* hybridization

Embryos were fixed overnight at 4°C with 4% formaldehyde in PBS. *In situ* hybridization was carried out as in (*Thisse and Thisse, 2008*) and representative embryos were imaged in 2:1 benzyl benzoate:benzyl alcohol (BBBA) with a Zeiss Axio Imager.Z1 microscope. When genotyping was necessary, genomic DNA was generated after imaging as in (*Meeker et al., 2007*) and used in the described genotyping assays.

### Nodal inhibitor drug

The Nodal inhibitor SB-505124 (S4696, Sigma-Aldrich) (*DaCosta Byfield et al., 2004*; *Fan et al., 2007*; *Hagos et al., 2007*; *Hagos and Dougan, 2007*; *van Boxtel et al., 2015*; *Vogt et al., 2011*) was dissolved in DMSO to a generate a stock at 10 mM and stored at 4°C. Fresh dilutions were made the same day experiments were carried out. Some batch-to-batch variability occurred, as well as a slight decrease in efficacy of the stock over time. The precise rescuing drug concentration must therefore be empirically determined for each stock of SB-505124.

## Acknowledgements

We thank María Almuedo-Castillo, Jeffrey Farrell, Hans Meinhardt, and Patrick Müller for insightful comments and discussions, Laila Akhmetova and Julien Dubrulle for experimental support, Antonius van Boxtel, John Chesebro, and Caroline Hill for sharing their pSmad2 immunofluorescence protocol, and Joseph Zinski and Mary Mullins for immunofluorescence imaging advice. Funding was provided by the National Science Foundation (KWR), the Arnold and Mabel Beckman Foundation (NDL), the American Cancer Society (JAG), and the National Institutes of Health (AFS GM056211, AP K99 HD076935). JKJ was supported by an NIH Director's Pioneer Award (DP1 GM105378) and NIH R01 GM088040. JKJ is a consultant for Horizon Discovery. JKJ has financial interests in Beacon Genomics, Editas Medicine, Poseida Therapeutics, and Transposagen Biopharmaceuticals. JKJ's interests were reviewed and are managed by Massachusetts General Hospital and Partners HealthCare in accordance with their conflict of interest policies.

## Additional information

### Competing interests

J Keith Joung: JKJ is a consultant for Horizon Discovery. JKJ has financial interests in Beacon Genomics, Editas Medicine, Poseida Therapeutics, and Transposagen Biopharmaceuticals. JKJ's interests were reviewed and are managed by Massachusetts General Hospital and Partners HealthCare in

accordance with their conflict of interest policies. The other authors declare that no competing interests exist.

## Funding

| Funder | Grant reference number | Author |
| --- | --- | --- |
| National Science Foundation | Graduate Research Fellowship | Katherine W Rogers |
| National Institutes of Health | AFS-GM056211 | Alexander F Schier Katherine W Rogers |
| Arnold and Mabel Beckman Foundation | Postdoctoral Fellowship | Nathan D Lord |
| American Cancer Society | Postdoctoral Fellowship | James A Gagnon |
| National Institutes of Health | AP-K99 -HD076935 | Andrea Pauli |
| National Institutes of Health | JKJ-R01-GM088040 | J Keith Joung |
| National Institutes of Health | JKJ-DP1-GM105378 | J Keith Joung |

The funders had no role in study design, data collection and interpretation, or the decision to submit the work for publication.

## Author contributions
Katherine W Rogers, Conceptualization, Formal analysis, Investigation, Visualization, Methodology, Writing—original draft, Writing—review and editing; Nathan D Lord, Conceptualization, Software, Formal analysis, Investigation, Visualization, Methodology, Writing—original draft, Writing—review and editing; James A Gagnon, Andrea Pauli, Resources, Methodology, Writing—review and editing; Steven Zimmerman, Deepak Reyon, Shengdar Q Tsai, J Keith Joung, Resources, Methodology; Deniz C Aksel, Investigation; Alexander F Schier, Conceptualization, Resources, Supervision, Funding acquisition, Writing—original draft, Project administration, Writing—review and editing

## Author ORCIDs
Katherine W Rogers http://orcid.org/0000-0001-5700-2662
Nathan D Lord http://orcid.org/0000-0001-9553-2779
Andrea Pauli https://orcid.org/0000-0001-9646-2303
Alexander F Schier https://orcid.org/0000-0001-7645-5325

## Ethics
Animal experimentation: This study was performed in strict accordance with the recommendations in the Guide for the Care and Use of Laboratory Animals of the National Institutes of Health. All of the animals were handled according to approved institutional animal care and use committee (IACUC) protocols (#25-08) of Harvard University.

## Decision letter and Author response
Decision letter https://doi.org/10.7554/eLife.28785.027
Author response https://doi.org/10.7554/eLife.28785.028

## Additional files

### Supplementary files
• Source code 1. MATLAB image processing scripts.
DOI: https://doi.org/10.7554/eLife.28785.023

• Transparent reporting form
DOI: https://doi.org/10.7554/eLife.28785.024

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
