## [Decision Letter]

Thank you for submitting your article "Nodal patterning without Lefty inhibitory feedback is functional but fragile" for consideration by *eLife*. Your article has been reviewed by three peer reviewers, and the evaluation has been overseen by a Reviewing Editor and Marianne Bronner as the Senior Editor. The reviewers have opted to remain anonymous.

The reviewers have discussed the reviews with one another and the Reviewing Editor has drafted this decision to help you prepare a revised submission.

Summary:

In this manuscript, Rogers et al. devise an interesting test of the relevance of negative feedback in a signaling circuit during early zebrafish development. Nodal signaling, which is activated by the ligands Cyclops and Squint, is kept under control by two secreted feedback inhibitors, Lefty 1 and Lefty 2. Here, the authors investigate the relevance of this regulatory interaction by assessing the effect of exogenous Nodal inhibitors in embryos lacking Lefty 1/2. One of their most remarkable results is that these mutants can be fully rescued by addition of a chemical inhibitor of Nodal signalling (at the just-right dose, but in a non-spatially restricted manner). This result clearly shows that signaling-dependent feedback inhibition is not required for normal development under laboratory conditions. The Lefty proteins are not needed to shut down Nodal signaling through increased production, nor are they absolutely essential for mesendodermal patterning that is compatible with viability. The elegance of the reaction-diffusion model also seems to be dispensable under these circumstances. The authors then suggest that feedback inhibition may nevertheless be important to buffer against fluctuations of signaling activity. The latter argument makes sense but is incompletely investigated; some of the suggested revisions may help to strengthen this argument.

Overall, this is a very strong and well-written manuscript, leading to a reconsideration of the concept of feedback loops as exemplified by the roles of Nodal ligands and their induced inhibitors Lefty 1/2. This work will undoubtedly generate a lot of discussion in the field. The large amount of data and the careful performance of the experiments will be greatly appreciated by those who want to carefully evaluate the data in light of various models for pattern formation. Consideration of the issues listed below is likely to strengthen the work and enhance its impact.

Essential revisions:

1) The authors suggest that there is a sweet spot of signaling activity that is required for normal development. This is defined by the dose of inhibitor that is required to support development of the double mutant. An alternative (quantitative) measure of signaling activity (e.g. western blot with anti-pSMAD) in chemically and genetically rescued embryos would strengthen the authors' argument. This would allow them to estimate the tolerance to variation in signaling activity, which is relevant to their argument that the normal function of feedback is to ensure reliable signaling at the right level.

2) The authors predict that signaling activity would be variable in the absence of feedback. Is it possible to strengthen this argument by measuring embryo to embryo variation among a population of drug-rescued Lefty double mutants (no feedback) with a quantitative measure as suggested above? Would such variation fall outside the tolerance measured in the previous experiment?

3) One possible extension of the finding described is that the spatial requirement for Nodal signalling is not very stringent. Have the authors attempted a converse rescue experiment, i.e. to activate Nodal signaling uniformly, either with a drug or RNA injection in a *cyclops;squint* double mutant. Would they expect a just-right level of global signaling that rescues development?

4) Regarding the statement "Thus, an ectopic, Nodal-independent source of Lefty at the animal pole can replace endogenous, Nodal-induced Lefty at the margin": Is this confirmed by fate-mapping (with Alexa 488-dextran) of the transplanted, Lefty-expressing cells? Figure 5 shows the position of cells shortly after transplantation, but it is quite possible that the transplanted cells move down into the margin during epiboly/gastrulation, providing a more localized source of exogenous Lefty.

5) It is somewhat surprising that there is not a graded effect of loss of lefty alleles, i.e. that the embryonic phenotypes and response readouts (in situs and pSmad2) do not show a graded response, and that loss of one, two or three alleles of Lefty1/2 seem to be equivalent. Is it correct that there is no dose response? What are the phenotypes of the genetic combinations with just one functional allele?

6) In the Discussion, "While Lefty is not essential for viability per se" is an overstatement. The lefty double mutants are not viable. Although they can be experimentally rescued by drug or resupply of Lefty, this does not alter the requirement for functional Lefty during development (whether provided by expression or downstream pathway inhibition), as demonstrated by the double mutant phenotype. It is important in the interpretation of the results and the discussion of the results to uncouple the concept of dynamic feedback inhibition with the concept of the requirement of Lefty to be in the game, at least to modulate Nodal activity levels for viability.

---

## [Author Response]

Essential revisions:1) The authors suggest that there is a sweet spot of signaling activity that is required for normal development. This is defined by the dose of inhibitor that is required to support development of the double mutant. An alternative (quantitative) measure of signaling activity (e.g. western blot with anti-pSMAD) in chemically and genetically rescued embryos would strengthen the authors' argument. This would allow them to estimate the tolerance to variation in signaling activity, which is relevant to their argument that the normal function of feedback is to ensure reliable signaling at the right level.

We thank the reviewers for this helpful suggestion. We have performed additional experiments to address this point: To examine the tolerance to variation in signaling activity, we used pSmad2 immunofluorescence to measure Nodal signaling in *lft* double mutants exposed to rescuing, sub-rescuing, and excessive doses of inhibitor drug (new Figure 6—figure supplement 4). Mutants exposed to sub-rescuing doses had expanded Nodal signaling gradients compared to rescued mutants. In contrast, mutants exposed to excessive doses exhibited pSmad2 gradients with diminished amplitude. Concurrently, treatment with excessive or insufficient Nodal inhibitor doses resulted in morphological phenotypes consistent with partial Nodal loss of function and gain of function, respectively. Our new results therefore suggest upper and lower bounds of tolerance to signaling alterations, and support the argument that mechanisms to maintain signaling within these bounds are important for patterning. We have updated the main text accordingly.

2) The authors predict that signaling activity would be variable in the absence of feedback. Is it possible to strengthen this argument by measuring embryo to embryo variation among a population of drug-rescued Lefty double mutants (no feedback) with a quantitative measure as suggested above? Would such variation fall outside the tolerance measured in the previous experiment?

We thank the reviewers for raising this possibility. We have performed additional experiments to address this point: Using pSmad2 immunofluorescence, we observed similar levels of signaling gradient variability in wild type and drug-rescued *lft* mutants (Author response image 1). This result provides further support for the remarkable ability of the inhibitory drug to rescue *lft* mutants to a wild type phenotype. Although this result does not provide additional evidence for a role of inhibitory feedback in reducing variation, there are technical limitations to the sensitivity of this experiment using currently available methods. To control for technical variability in staining, we followed the accepted practice of normalizing each embryo’s gradient to its total intensity. This approach is well suited to analyze variability in gradient shape but it normalizes biological variability in gradient amplitude. It is therefore conceivable that there might be differences in amplitude variability between wild type and rescued mutants. In future studies we will attempt to develop more quantitative measures of absolute staining intensities, a challenge that has also been highlighted as problematic in previous studies (e.g. Bollenbach et al. 2008, Gregor et al. 2007).

**Author response image 1. respfig1:** Nodal signaling gradients in wild type and rescued *lefty* double mutants have comparable variability. All data were derived from quantitative analysis of 50% epiboly embryos stained for pSmad2 as described in the Materials and methods section. Nuclear intensities from each embryo were normalized to the total gradient intensity (i.e., the integral of the activity gradient) to control for technical variability in antibody staining. A) Normalized Nodal activity gradients in wild type embryos. Twelve gradients derived from six embryos (i.e., gradients from the left and right sides of embryo cross-sections) are plotted in light blue. Each gradient is estimated with a sliding window average of average nuclear pSmad2 staining intensity. The mean gradient shape was calculated by taking a sliding window average of pooled data (dark blue). B) Normalized Nodal activity gradients in *lft* double mutants rescued by exposure to 2 µM SB-505124 starting at the 8-cell stage. Twelve gradients were derived from six embryos (light red), and the mean gradient was estimated with a sliding window average of pooled data (dark red). C) Mean wild type (blue) and rescued *lefty* double mutant (red) gradients are plotted for comparison. Error bars denote 1 standard deviation of data within 8 µm bins. Although the signaling gradient shapes differ between wild type and drug-rescued *lft* double mutant embryos, they appear to have similar variability. Note, however, that the accepted practice of normalizing by total intensity may diminish true biological variability in gradient amplitude.

3) One possible extension of the finding described is that the spatial requirement for Nodal signalling is not very stringent. Have the authors attempted a converse rescue experiment, i.e. to activate Nodal signaling uniformly, either with a drug or RNA injection in a cyclops;squint double mutant. Would they expect a just-right level of global signaling that rescues development?

It is well established that ectopic activation of Nodal signaling leads to ectopic mesendoderm induction in wild type embryos, but the reviewers are correct that it has not been tested stringently whether broad activation of the pathway can rescue Nodal signaling mutants. The genetics of creating *cyc;sqt* double mutants (1/16 embryos are double mutants) makes such an experiment extremely challenging. We have instead extended the experiments performed in Gritsman et al. 1999, in which the Nodal co-receptor mutant MZ*oep*, which is completely devoid of Nodal signaling, is injected with activin mRNA. Activin bypasses the requirement for *oep* and is able to activate the Nodal signaling pathway in a dose-dependent manner (Dubrulle et al. 2015). We injected different amounts of *activin* mRNA into MZ*oep* embryos at the one- to two-cell stage and observed partial rescue of some phenotypes, such as loss of trunk mesendoderm and, in one rare instance, rescue of cyclopia (new Figure 5—figure supplement 3). However, these embryos had severe morphological defects, in stark contrast to rescued *lft* double mutants (Figure 6, Figure 6—figure supplement 1 and Figure 6—figure supplement 2). We therefore conclude that the spatial requirements for Nodal activity are much stricter than for Lefty. We have updated the paper accordingly.

4) Regarding the statement "Thus, an ectopic, Nodal-independent source of Lefty at the animal pole can replace endogenous, Nodal-induced Lefty at the margin": Is this confirmed by fate-mapping (with Alexa 488-dextran) of the transplanted, Lefty-expressing cells? Figure 5 shows the position of cells shortly after transplantation, but it is quite possible that the transplanted cells move down into the margin during epiboly/gastrulation, providing a more localized source of exogenous Lefty.

We thank the reviewers for requesting this control experiment. Previous studies (e.g. Dubrulle et al. 2015, Chen et al. 2001, Müller et al. 2012) already showed that transplanted cells tend to remain at the animal pole. To directly determine the location of transplanted cells in our experiments, we injected Alexa 488-dextran, membrane RFP, and *lft1* mRNA into *lft* double mutant embryos and transplanted cells from these donors into membrane RFP-expressing host *lft* double mutants at sphere stage, as in Figure 5. We then imaged host embryos for 4.5 hours post-transplantation, covering the stages during which the majority of germ layer patterning occurs (new Figure 5—figure supplement 1). We found that transplanted cells tended to remain at the animal pole and later frequently localized to the head, consistent with an animal pole localization at earlier stages (new Figure 5—figure supplement 2). Most importantly in the context of the paper, the transplanted cells represent a radically different spatial source of Lefty compared to the endogenous source at the margin.

5) It is somewhat surprising that there is not a graded effect of loss of lefty alleles, i.e. that the embryonic phenotypes and response readouts (in situs and pSmad2) do not show a graded response, and that loss of one, two or three alleles of Lefty1/2 seem to be equivalent. Is it correct that there is no dose response? What are the phenotypes of the genetic combinations with just one functional allele?

We thank the reviewers for pointing out this interesting observation. Embryos with even a single functional *lft* allele appeared largely morphologically normal (Figure 1), and gave rise to fertile adult fish, which we incrossed to obtain double homozygous embryos for several of the experiments in this study. These results are consistent with the feedback induction of *lfts* by Nodal signaling and the previous observation that *lfts* are highly sensitive Nodal target genes – they have the highest transcription rates of all targets in response to Nodal signaling, and *lft* transcripts are among the least stable (Dubrulle et al. 2015). Moreover, we also observed high sensitivity of *lft2* expression to alterations in Nodal levels (Figure 7). We therefore speculate that robustness to *lft* allele dosage is the result of compensation from the remaining alleles, which may be induced rapidly in response to slightly increased Nodal signaling. We note, however, that subtler patterning defects may occur in *lft*-reduced embryos; for example, we did observe heart patterning defects in some *lft1^-/-^* mutants (Figure 1—figure supplement 1). In the future it will be interesting to more closely assess the patterning of Nodal-dependent structures in both *lft1* and *lft2* mutants in greater detail (e.g. lateral plate mesoderm, heart, endoderm, left-right positioning of organs, etc.).

6) In the Discussion, "While Lefty is not essential for viability per se" is an overstatement. The lefty double mutants are not viable. Although they can be experimentally rescued by drug or resupply of Lefty, this does not alter the requirement for functional Lefty during development (whether provided by expression or downstream pathway inhibition), as demonstrated by the double mutant phenotype. It is important in the interpretation of the results and the discussion of the results to uncouple the concept of dynamic feedback inhibition with the concept of the requirement of Lefty to be in the game, at least to modulate Nodal activity levels for viability.

We agree with the reviewers and removed this statement to change the sentence to: “The ability to dynamically adjust pathway activity may allow the embryo to create reliable patterns in the face of signaling fluctuations and uncertain environmental conditions”.